# The RPA-binding domain and the KKRK motif in Rad26ATRIP cooperate at the perturbed DNA replication fork for initiating checkpoint signalling

Yong-jie Xu[1]*, Anmin Gao[2], Kamal Dev[1], Yuyuan Zheng[2], Mashael Y. Alyahya[1], Sairam Pasam[1], Guramrit Kaur[1], Chun Zhou[2]

1 Department of Pharmacology and Toxicology, Boonshoft School of Medicine, Wright State University, Dayton, Ohio, United States of America, 2 School of Public Health and Sir Run Run Shaw Hospital, Zhejiang University School of Medicine, Hangzhou, China

* yong-jie.xu@wright.edu

## Abstract

Rad26 is the homolog of human ATRIP and budding yeast Ddc2 in *Schizosaccharomyces pombe*. Like ATRIP and Ddc2, Rad26 works with Rad3ATR/Mec1 to initiate checkpoint signalling in response to perturbed DNA replication and various types of DNA damage. To better understand the checkpoint initiation mechanism in fission yeast, we carried out genetic and biochemical analyses on the N-terminus of Rad26. Although Rad26 homologs do not share much sequence similarity, we demonstrate that, like ATRIP and Ddc2, Rad26 possesses a replication protein A (RPA) binding domain (RBD) in its N-terminus, suggesting a highly conserved mechanism. Elimination of the RBD in Rad26, however, only moderately affects the checkpoint signalling and cellular resistance to genotoxins. Rad26 has a short KKRK sequence in the N-terminal region, a motif conserved in Ddc2 that binds DNA and is crucial for the checkpoint function in budding yeast. Mutations of this motif in Rad26 cause only a minor defect in the checkpoint. However, simultaneous mutations of the RBD and the KKRK motif nearly eliminate the Rad3ATR kinase signalling at the perturbed replication fork. This suggests that the two functional units of Rad26 cooperate to initiate the DNA replication checkpoint. On the contrary, the simultaneous mutations of Rad26 only moderately or minimally sensitize the cell to different types of DNA damage. We hypothesize that the checkpoint initiation at the DNA damage site in fission yeast may follow a different mechanism that depends less on the two functional units of Rad26.

## Author summary

Rad26ATRIP is a checkpoint sensor protein in the fission yeast *S. pombe*. Like its homologous proteins in other model organisms, it collaborates with the

**Data availability statement:** All relevant data are in the manuscript and its Supporting information files.

**Funding:** This study was supported by NIH/ National Institute of General Medical Sciences, grant R35GM144307 to YjX and the National Natural Science Foundation of China, grant 32371250 to CZ. The funders had no role in study design, data collection and analysis, decision to publish, or preparation of the manuscript.

**Competing interests:** The authors have declared that no competing interests exist.

checkpoint sensor kinase Rad3$^{ATR}$ to trigger Rad3$^{ATR}$-mediated checkpoint signaling at perturbed DNA replication forks, activating the DNA replication checkpoint, or at the sites of DNA damage, initiating the DNA damage checkpoint. The current checkpoint model suggests that the binding to replication protein A (RPA) by Rad26$^{ATRIP}$ is required for checkpoint initiation. We demonstrate, unexpectedly, that removing the RPA-binding domain (RBD) in the N-terminus of Rad26$^{ATRIP}$ only moderately attenuates the Rad3$^{ATR}$ checkpoint signalling in fission yeast. However, when both the RBD and a KKRK motif in Rad26$^{ATRIP}$ are removed, the Rad3$^{ATR}$ checkpoint signalling at the replication fork is nearly abolished, indicating a cooperative mechanism between the RBD and the KKRK motif at the perturbed DNA replication fork. Interestingly, the simultaneous removal of the RBD and the KKRK motif has a minor or moderate effect on cellular resistance to DNA-damaging agents, indicating that checkpoint initiation at DNA damage sites is less dependent on these two functional units of Rad26$^{ATRIP}$.

## Introduction

Genome integrity is constantly challenged by numerous cellular and environmental factors that damage DNA or disrupt the accurate transmission of heritable information to the next generation [1,2]. DNA damage is also a cornerstone of cancer therapy. To maintain genome integrity, eukaryotes have evolved several dedicated mechanisms, including high-fidelity DNA replication, repair pathways of various types of DNA damage and replication errors, and the cell-cycle checkpoint system. The checkpoint monitors the cellular processes of DNA replication and repair and coordinates their activities with cell cycle progression. Consistent with its importance in genome maintenance, the checkpoint system is highly conserved in eukaryotes. Defects of the checkpoint pathways cause genome instability, a driving force of cancer initiation and progression.

The current checkpoint model suggests that a conserved set of sensor proteins assembles at the perturbed replication fork or DNA damage site to initiate the checkpoint signaling [3]. The checkpoint signaling, once initiated, is relayed through mediator proteins to the effector kinases, such as Chk1 and Chk2, to spread the signal throughout the cell. In mammalian cells, ataxia–telangiectasia mutated (ATM) activates the checkpoint mainly through Chk2 in response to double-strand breaks (DSBs), whereas ATR (ATM and Rad3-related) activates the checkpoint mainly through Chk1 in response to perturbed replication or various other types of DNA damage. For the ATR-initiated checkpoint signaling, replication protein A (RPA) is believed to play a critical role [4]. RPA is a heterotrimer complex highly conserved in eukaryotes [5]. It binds to the single-stranded DNA (ssDNA) associated with the fork or DNA damage sites. The RPA-coated ssDNA serves as a platform for the assembly of checkpoint sensor proteins, including ATR [4,6,7]. The N-terminal F-domain of Rpa1, the large subunit of RPA, binds to ATRIP, the binding partner of ATR, which recruits the ATR-ATRIP complex to initiate the checkpoint signaling in the DNA

replication checkpoint (DRC) and the DNA damage checkpoint (DDC) pathways [8]. The RPA-ssDNA platform also promotes the loading of Rad9-Rad1-Hus1 (9-1-1) checkpoint clamp at the 5' end of the ssDNA/dsDNA junction [7,9,10]. The loaded 9-1-1, upon phosphorylation by ATR, recruits more checkpoint proteins, such as the adaptor protein TopBP1 with a so-called ATR activation domain [11]. The recruitment of TopBP1 can therefore stimulate and enhance the ATR kinase signaling. Like TopBP1, ETAA1 can also activate ATR both *in vitro* and *in vivo* [12], similar to the budding yeast Ddc1, Dna2, and Dpb11 that activate Mec1$^{ATR}$ [13–15].

We have been investigating the checkpoint initiation mechanisms in *S. pombe*, an established yeast model for studying the cellular mechanisms that are conserved in higher eukaryotes. In fission yeast, Rad3$^{ATR}$ and its cofactor Rad26$^{ATRIP}$ mediate most of the checkpoint functions, whereas Tel1$^{ATM}$ contributes minimally to the checkpoint [16]. Although the DRC and DDC pathways in *S. pombe* are all initiated by Rad3-Rad26, they are mediated by Cds1$^{CHK2/Rad53}$ and Chk1, separately [16]. This promotes an unambiguous description of the checkpoint initiation mechanisms. We have recently screened the *ssb1* gene, which encodes the large subunit of RPA in fission yeast, looking for checkpoint mutants that are sensitive to the replication stress induced by hydroxyurea (HU) or DNA-damaging agent methyl methane sulfonate (MMS) [17]. This extensive screen, particularly the F-domain of Ssb1, identified 25 mutants that are sensitive to HU and other genotoxins. However, none of the screened mutants has a defective DDC, and only two mutants, particularly *ssb1–1*, have a partial defect in the DRC pathway. An early genetic study identified a series of mutants of *rfa1*, which encodes the large subunit of RPA in budding yeast [18]. Among these mutants, *rfa1-t11* with a K45E mutation in the F-domain of Rfa1 has been shown to have a checkpoint defect [4]. A recent biochemical and structural study has shown, however, that the K45 residue does not directly participate in the binding to Ddc2$^{ATRIP}$ in budding yeast [19]. When we examined the interaction of Rad26$^{ATRIP}$ with the Ssb1 by co-immunoprecipitation (co-IP) in the newly screened *ssb1–1* mutant in fission yeast, we found that the mutation did not affect the binding (see Fig AA in S2 File). This result echoes what is observed in the budding yeast *rfa1-t11* mutant [20,21] and raises a concern about the current model of the ATR checkpoint initiation mechanism. One explanation is that the checkpoint function of Ssb1 is masked by its essential function in cell growth. Since Rad26$^{ATRIP}$ is not required for cell growth, we then investigated in this study whether Rad26 interacts with RPA and, if it does, how it contributes to checkpoint initiation in fission yeast.

By *in vivo* and *in vitro* studies, we show that, like ATRIP and Ddc2, the N-terminus of Rad26 possesses an RPA-binding domain (RBD) that binds directly to the N-terminal F-domain of Ssb1. However, elimination of the RBD in Rad26 only moderately affects the checkpoint and the cellular resistance to HU and MMS. We then investigated the KKRK motif conserved in budding yeast Ddc2 and found that simultaneous mutations of this motif and the RBD almost eliminate the Rad3$^{ATR}$ kinase signalling at the HU-treated forks, suggesting that two functional units of Rad26 cooperate at the fork for checkpoint initiation. Interestingly, the simultaneous mutations did not significantly sensitize *S. pombe* to various types of DNA damage. We propose that the checkpoint initiation at the DNA damage site may involve a different mechanism that needs further study.

## Results

**The N-terminal region of Rad26$^{ATRIP}$ promotes Rad3$^{ATR}$ kinase signalling.** The fission yeast Rad26, like human ATRIP and budding yeast Ddc2, binds to Rad3$^{ATR}$ and promotes Rad3 kinase signalling in both the DRC and the DDC pathways [16,22]. In the DRC pathway, Rad3-Rad26 activates the effector kinase Cds1$^{CHK2}$ at the perturbed fork via the mediator protein Mrc1$^{Claspin}$ [23]. At the DNA damage site, Rad3-Rad26 activates the effector kinase Chk1 in the DDC pathway [24–26]. Mutants of the DRC and the DDC pathways are sensitive to replication stress and DNA damage. Although it is known that Rad26 is required for checkpoint initiation, the molecular mechanisms are not fully understood. Part of the reason is that the homologous proteins do not share much sequence similarity. Studies in other model systems suggest that Rad26 binds to the F-domain of Ssb1 (Fig AA in S2 File), the large subunit of RPA in *S. pombe*, which recruits the sensor kinase Rad3 to the fork or DNA damage site to initiate checkpoint signalling by phosphorylating Mrc1$^{Claspin}$, Cds1$^{Chk2}$, Chk1,

and other target proteins [4,16]. In a previous study, we investigated this mechanism by screening the checkpoint mutants of *ssb1* [17]. Although the screen was extensive, none of the identified mutants had a defect in the DDC pathway, and only two mutants, particularly *ssb1–1*, showed a partial defect in the DRC pathway. Further, our recent co-IP experiment showed that the interaction between Rad26 and Ssb1 is unaffected in the *ssb1–1* mutant (Fig AB in S2 File). This raises a concern about the current checkpoint initiation mechanism. One explanation is that the essential function of Ssb1 in cell growth prevents the identification of a non-lethal mutant with a checkpoint defect. We therefore investigated Rad26.

Although not conserved in primary amino acid (aa) sequences (FigBA in S2 File), Alphafold prediction showed that Rad26 shares a similar 3D structure with Ddc2 and ATRIP (Fig BB in S2 File). The C-terminal domain (CTD) is highly structured, which is believed to bind to Rad3 (Fig 1A). Also conserved is the N-terminal coiled-coil domain (CCD), which likely promotes the dimerization of the Rad26-Rad3 complex. The KKRK motif required for the checkpoint functions of Ddc2 is conserved in Rad26, but not in ATRIP. The N-terminal ~70 aa are less structured, except for the presence of a α-helix (Fig 1A). Several residues in the N-terminus of Ddc2 have been identified previously that are involved in the binding to the F domain of Rfa1 (Fig BA in S2 File, blue dots) [19]. Some of them are conserved in Rad26. To see whether the N-terminal region is required for the checkpoint functions of Rad26, we engineered a SpeI site with a silent mutation right after the 207th aa (Fig BB in S2 File, red arrow). Random mutations were generated by PCR in the 1–206 aa region to screen for mutations that sensitize the cells to HU or MMS. This preliminary study found that mutation of the F18 residue sensitizes the cells, particularly to HU (Fig BC in S2 File), indicating that the N-terminus of Rad26 may contribute to the checkpoint, like its counterparts in other model organisms.

We then made a series of deletions of the Rad26 N-terminal region and expressed them with an N-terminal HA epitope in Δ*rad26* cells under the control of its native promoter. Western blotting showed that all mutants, except the Δ162, were expressed at levels comparable or slightly lower than the full-length Rad26 (Fig 1B). When the sensitivities to HU and MMS of these mutants were examined by spot assay (Fig 1C), we found that while the Δ10 deletion did not sensitize the cells, the Δ20 and Δ30 deletions moderately sensitized the cells to HU and MMS. Further deletions of Δ50 and Δ70 significantly sensitized the cells, particularly HU, although the protein levels were slightly lower (Fig 1B). This result suggests that the first 20–50 aa are important for the checkpoint, particularly the DRC pathway. Since the protein level of Δ162 protein was significantly lower (Fig 1B, arrow), we overexpressed Δ162 and even Δ220 with only the CTD domain under a strong *nmt1* promoter. Overexpression of the two mutants slowed cell growth but mildly improved drug resistance (Fig CA and CB in S2 File). However, Rad3-specific phosphorylation of Mrc1 and Chk1 was not detected in the presence of HU or MMS (Fig CC in S2 File). The low level of Mrc1 in HU-treated cells is likely due to the cell cycle delay caused by overexpression of Rad26-CTD. This result suggests that although the overexpressed CTD domain can weakly promote cell survival in the presence of HU and MMS, the N-terminal region of Rad26 may facilitate the kinase signalling of Rad3, particularly at the DNA damage sites induced by MMS, which mainly occurs at G2, the longest cell cycle time in fission yeast.

**The N-terminal region of Rad26^ATRIP promotes Rad3^ATR site-specific kinase signalling.** When DNA replication is perturbed in fission yeast, Rad3^ATR phosphorylates two TQ motifs (T645 and T653) in the middle of Mrc1^Claspin [23,27]. Phosphorylated Mrc1 then recruits Cds1 for Rad3 phosphorylation at Cds1-T11. Phosphorylated Cds1-T11 promotes homodimerization of Cds1 and autophosphorylation of Cds1-T328. Phosphorylation of T328 in the activation loop directly activates Cds1 [28], which mediates most of the biological functions of the DRC in *S. pombe*. At the DNA damage site, Rad3 phosphorylates Chk1-S345 to activate the DDC via the mediator Crb2^53 BP1/Rad9 [24–26,29]. To examine Rad3 kinase signalling in the Rad26 deletion mutants, we monitored the site-specific phosphorylation of Mrc1 and Chk1 by Rad3, the key molecular markers of the DRC and the DDC pathways, respectively. While Mrc1 phosphorylation was examined by phospho-specific antibody [27], the phosphorylation of Chk1 was examined by the mobility shift assay [24]. As shown in Fig 1D (upper panels), when treated with HU, phosphorylation of Mrc1-T645 was significantly increased in wild-type cells. Since Mrc1 is specifically expressed during the G1/S phase and activated DRC promotes the expression, HU

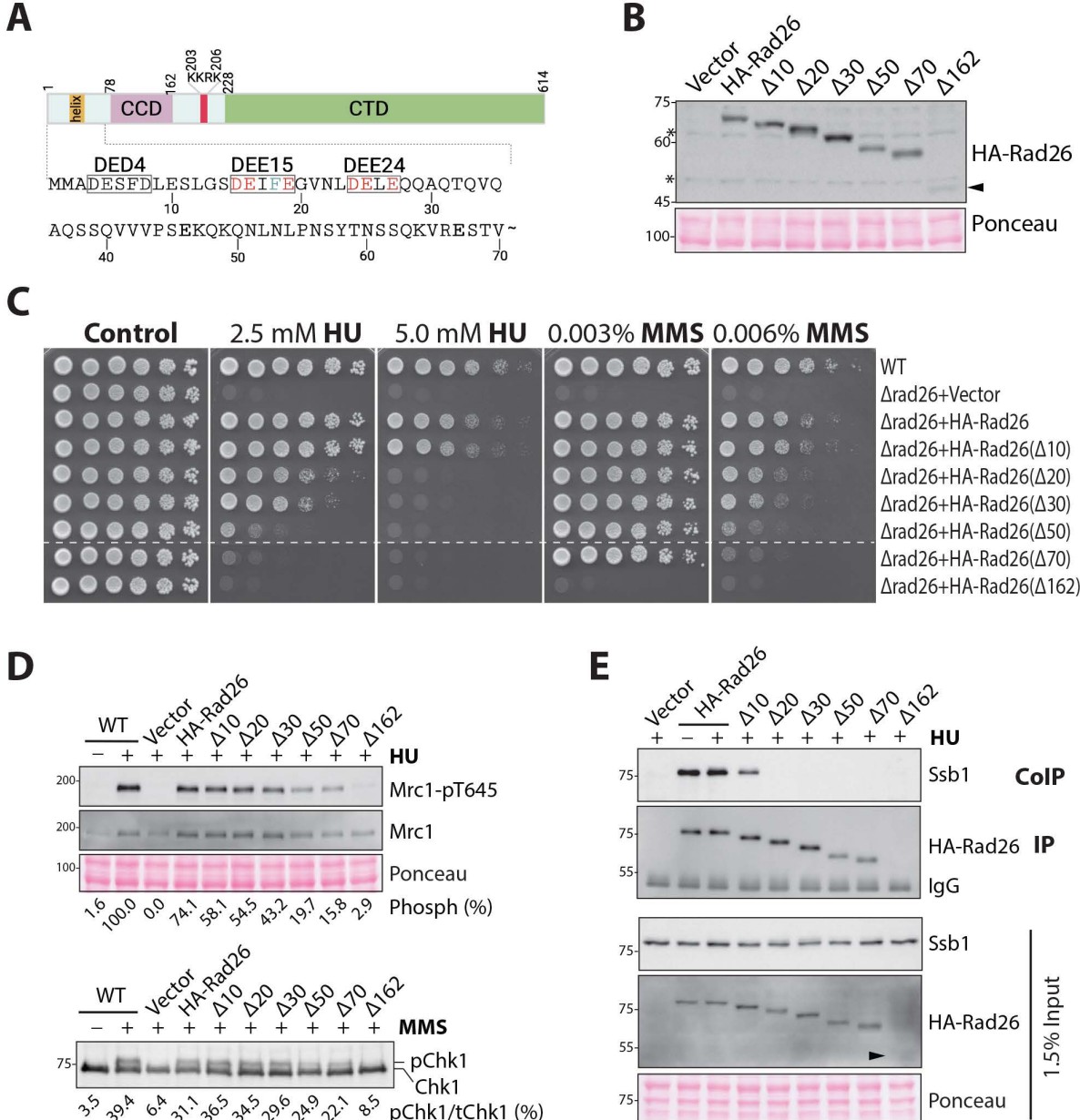

**Fig 1. The N-terminal region of Rad26 binds to RPA and promotes checkpoint signalling. (A)** Rad26 molecular architecture predicted by Alpha-Fold. The N-terminal 70 aa region is enlarged with three highlighted motifs: DED4, DEE15, and DEE24. An α-helix, the coil-coiled domain (CCD), the KKRK motif, and the C-terminal domain (CTD) are marked in their relative positions. **(B)** A series of deletions was made from the 1st amino acid in HA-Rad26 expressed on a vector in Δrad26 cells. The expression levels of the full-length and the mutant proteins were examined in whole cell lysate by Western blotting using α-HA antibody. Asterisks indicate cross-reacting materials, and the arrow marks the Δ162 protein. **(C)** Sensitivities of the cells expressing full-length and the indicated mutants of Rad26 were examined by spot assay. A series of five-fold dilutions of the logarithmically growing cells was spotted on YE6S plates or plates containing the HU or MMS at the indicated concentrations. The plates were incubated at 30°C for 3 days and then photographed. Wild-type and Δrad26 cells carrying an empty vector were used as the controls. The dashed line indicates discontinuity. **(D)** Attenuation of the Rad3 kinase signalling in the DRC and the DDC pathways in the deletion mutants. Phosphorylation of Mrc1 by Rad3 was examined using the antibody against Mrc1-pT645 residue in the presence (+) or absence (-) of 15 mM HU for 3 h (top panel). The membrane was stripped and reblotted with the antibody against Mrc1 (2nd panel from the top). A section of the Ponceau S-stained membrane was shown as the loading control (3rd panel from the top). Quantitation results of the band intensities in the top panel are shown in percentages relative to the HU-treated wild-type cells. Bottom panel: Wild-type and the cells with the indicated Rad26 deletions were treated with (+) or without (-) 0.009% MMS for 90 min. Chk1 phosphorylation by Rad3 in the DDC pathway was examined by mobility shift using α-HA antibody to detect the Chk1 [24]. The intensities of the bands were measured and shown

at the bottom in ratios of pChk1 over total Chk1. **(E)** Deletion of the first 20 or 30 aa in the Rad26 N-terminus eliminated the binding to RPA. HA-Rad26 and the indicated mutants were expressed on a vector in Δ*rad26* cells. Cells with an empty vector were included as a control. HA-Rad26 was IPed using α-HA antibody bound on magnetic beads from the whole cell lysates made from the cells treated with (+) or without (-) HU. The membrane was first blotted with α-Ssb1 antibody to reveal the co-IPed Ssb1 (top panel). After stripping, the membrane was reblotted with α-HA antibody to visualize the IPed HA-Rad26 (2ⁿᵈ panel from the top). The lower three panels are 1.5% inputs of the co-IP experiment.

treatment also increases the protein levels of Mrc1. Deletion of the first 30 aa in Rad26 moderately reduced the phosphorylation of Mrc1. Further deletions significantly reduced the phosphorylation, and the reduced levels are consistent with the increased HU sensitivity (Compare Fig 1D with 1C). Similar results were observed in Chk1 phosphorylation in the presence of MMS (Fig 1D, bottom panel), indicating that the N-terminal region of Rad26 can promote site-specific Rad3 signalling for cell survival.

**Identification of the RBD in Rad26 N-terminus.** We then investigated whether the checkpoint function of the Rad26 N-terminal region is mediated through interacting with RPA. By co-IP and Western blotting using an α-Ssb1 antibody [17], we found that Ssb1, the large subunit of RPA in fission yeast, can be co-IPed with Rad26. Since the deletion mutants were more sensitive to HU than MMS (Fig 1C), HU treatment was included in the co-IP experiment. We found that Rad26 binding to Ssb1 is independent of HU treatment (Fig 1E), suggesting a constitutive binding of Rad26 to RPA, unlike an early observation in budding yeast [30]. The binding was significantly reduced in Δ10, although the deletion did not sensitize the cells to HU and MMS. Further deletions eliminated the binding. Since the phosphorylation of Mrc1 and Chk1 and drug resistance were only moderately reduced in Δ20 and Δ30 mutants, this discrepancy may be caused by the low sensitivity of the α-Ssb1 antibody used for the Western blotting. To address this issue, we tagged Ssb1 with 5xflag at the C-terminus at the genomic locus in Δ*rad26* cells so that the co-IPed Ssb1 can be detected with α-flag monoclonal antibody. Drug sensitivity assay showed the tagging did not affect the function of Ssb1 (Compare Fig DA in S2 File with Fig 1C). The co-IP results showed that, although the α-flag antibody has a higher background noise in the Western blot, the Δ20 and Δ30 deletions reduced the co-IPed Ssb1 to the background level (Fig DB in S2 File). To rule out the possibility that the N-terminal HA tag of Rad26 may affect the binding, we migrated the HA epitope to the C-terminus. Since the Rad26 N-terminus has two methionines, we made the deletions from the 3ʳᵈ aa. Spot assay showed that the C-terminal tagging of Rad26 did not significantly affect the functions, and the mutant proteins behaved quite similarly to the N-terminally tagged Rad26 (Compare Fig 1C with Fig DA and DC in S2 File). The co-IP results showed that, like the results in Fig 1E, deletion of the first 20 or 30 aa eliminated the binding of Rad26 to Ssb1 (Fig DD in S2 File). Together, these results show that deletion of the first 30 aa in Rad26 abolishes the binding to Ssb1. Since the N-terminal tagging of Rad26 does not affect its checkpoint function and the interaction with Ssb1, the N-terminal-tagged Rad26 is used in the rest of this study.

We then confirmed the interaction of Rad26 with Ssb1 by reciprocal co-IP. Similar results were observed, although the levels of co-IPed Rad26 were much lower (compare Fig 1E with Fig EA in S2 File). We also examined whether the N-terminal deletions affect Rad26 interaction with Rad3 by co-IP (Fig EB in S2 File). As expected, the N-terminal deletions, particularly the Δ20 and Δ30, did not affect the interaction with Rad3. The lower levels of co-IPed Rad3 in Δ50, Δ70, and Δ162 mutants are likely due to the lower levels of the Rad26 mutant protein.

**Identification of the key residues in the RBD of Rad26.** Our preliminary data have uncovered that the F18 residue plays an important role in Rad26 checkpoint function (Fig BC in S2 File). As shown in Fig 1A, the N-terminus of Rad26 carries three DED/DEE motifs with either one or two hydrophobic residues. The F18 resides in the 2ⁿᵈ DEE motif. Since the N-terminal acidic residues of Ddc2 play an important role in the interaction with the F-domain of Rfa1 in budding yeast [19], we substituted the three acidic residues in each of the three DED/DEE motifs with alanine and examined the mutational effect on drug sensitivity and the binding to Ssb1. For simplicity, we named the three motifs as DED4, DEE15, and DEE24, respectively. Among the three DED/DEE motifs, simultaneous substitution of the three acidic residues in DEE24 significantly sensitized the cells to HU and MMS (Fig FA in S2 File). Combinational mutations of DED4 with DEE15, and

DEE15 with DEE24, also significantly increased the drug sensitivity. However, the mutation of DED4, DEE24, and the combined mutations of DED4,15 and DED4,15,24 significantly reduced the protein level (Fig FB in S2 File, left panels). On the contrary, the mutation of DEE15 and the combined mutation of DEE15,24 did not reduce the protein level, and therefore, their drug sensitivity is likely due to the mutation. We further mutated one hydrophobic residue within each of the three DED/DEE motifs to alanine. As shown in the lower part of Fig FA in S2 File, only the F18A mutation sensitized the cells to HU, although the sensitivity was minimal in MMS. Unlike the mutations of the DED/DEE motifs, mutations of the three hydrophobic residues, whether single or in combinations, did not affect the protein levels of Rad26 (Fig FB in S2 File, right panels). The mutational effects of Rad26 are summarized in Table D in S1 File.

We then examined Rad26 binding with Ssb1 in the mutants by co-IP (Fig FC in S2 File, left half). The results showed that while DED4 plays a less important role, DEE15 and DEE24 are required for the binding. The reduced protein level of the DED4 mutant may also contribute to the weak binding to Ssb1 (Fig FC in S2 File). Similar to the weak binding of the DED4 mutant, the Δ10 mutant also showed a weak binding to Ssb1 (Fig 1E). Among the three hydrophobic residues mutated, only F18 is crucial for the binding to RPA (Fig FC in S2 File, right half). We also mutated the other two acidic residues, E46 and E67, in the Δ10 background, and the results clearly showed that mutations of the two residues did not affect drug resistance (Fig GA in S2 File), the protein levels (Fig GB in S2 File), and Rad26 binding to Ssb1 (Fig GC in S2 File, right half).

We then combined the F18A mutation with the mutations of the DEE motifs. When combined with DEE15 and DEE24, it sensitized the cells to HU and MMS like the Δ30 mutant (Fig 2A, compare Δ30 with F18-DEE15,24). Western blotting showed that the protein level of this mutant was comparable to that of full-length and the Δ30 proteins (Fig 2B). Co-IP showed that all mutants lost the binding to Ssb1. The abolished binding in the DED4 mutant is likely due to the significantly reduced Rad26 level. Together, these results showed that the DEE15 motif, including F18, and the DEE24 motif within the first 30 aa in the RBD of Rad26 are the key residues for binding to Ssb1. Like the Δ30 deletion, the combined mutations of the key residues only moderately affected Rad3 kinase signalling and drug sensitivities (Fig 2A and below).

**Interdependent molecular interactions between Rad26-RBD and Ssb1-F.** The results described above showed that Rad26-RBD binds to Ssb1. However, it remains unclear whether the binding is direct or indirect via other factors. AlphaFold3 prediction [31] showed that the RBD of Rad26 (beige) directly interacts with the F domain of Ssb1 (teal) (Fig 3A). The acidic residues of DEE15 and DEE24 motifs extensively interact with Ssb1-F via charge-charge interactions with residues such as R46, R48, and R94 in Ssb1. A series of hydrophobic residues in Rad26-RBD, such as I21, V21, L23, L26, particularly F18, are in proximity to the hydrophobic residues M61, M91, I96, and I98 in Ssb1-F, suggesting extensive hydrophobic interactions. To investigate these interactions biochemically, we purified the recombinant F-domain of Ssb1 (1–122 aa), chemically synthesized the Rad26 N-terminal (11–40 aa) peptides, and measured their binding affinities by isothermal titration calorimetry (ITC) in a buffer containing 100 mM NaCl. As shown in Figs 3B and H in S2 File, when the wild-type Rad26 peptide was injected into the solution containing the purified Ssb1-F, a typical sigmoidal binding curve (red) was observed, indicating a single binding mode with the dissociation constant Kd of ~18.9 μM. When three other Rad26 peptides with alanine substitutions of F18 (green), D15-E16 (blue), and D24-E27 (brown), respectively, were used in the assay under similar conditions, all three peptides lost the binding activity (Figs 3B and HB-HD in S2 File). These data are quite similar to the co-IP results (Fig 2C), which showed that Rad26 directly binds to RPA via the interaction between Rad26-RBD and Ssb1-F domain, and the binding involves a series of interdependent polar and hydrophobic interactions.

**Elimination of the RBD has a minor or moderate effect on Rad3ATR kinase signalling.** The above-described results showed that, like ATRIP and Ddc2, the N-terminus of Rad26 contains an RBD that directly binds to RPA regardless of the HU treatment. However, robust Rad3 kinase signalling in the DRC and the DDC pathways was still observed in the mutants of F18-DEE15,24 or Δ30 lacking the RBD (Fig 1D). Consistent with robust Rad3 signalling, the mutant cells were only moderately sensitive to HU and mildly to MMS (Fig 1C). Similar results were also observed by other groups in

**A**

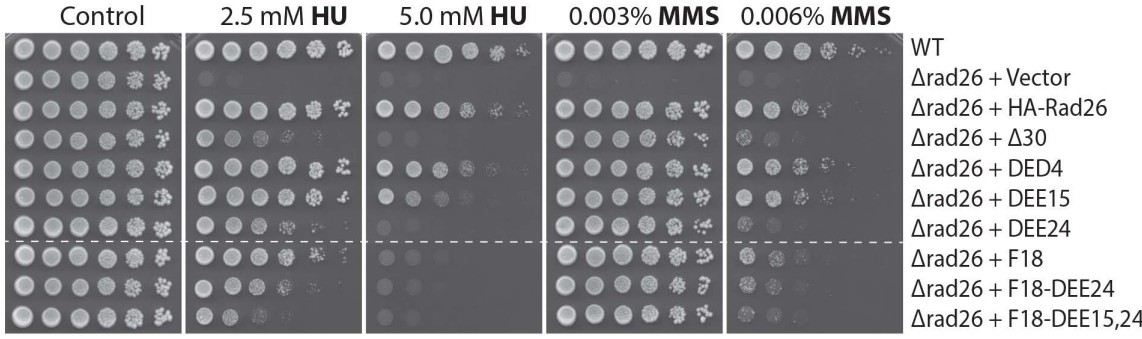

**B**

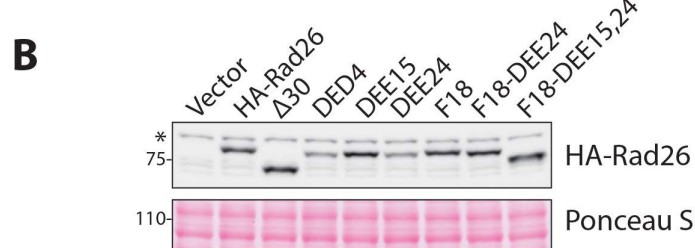

**C**

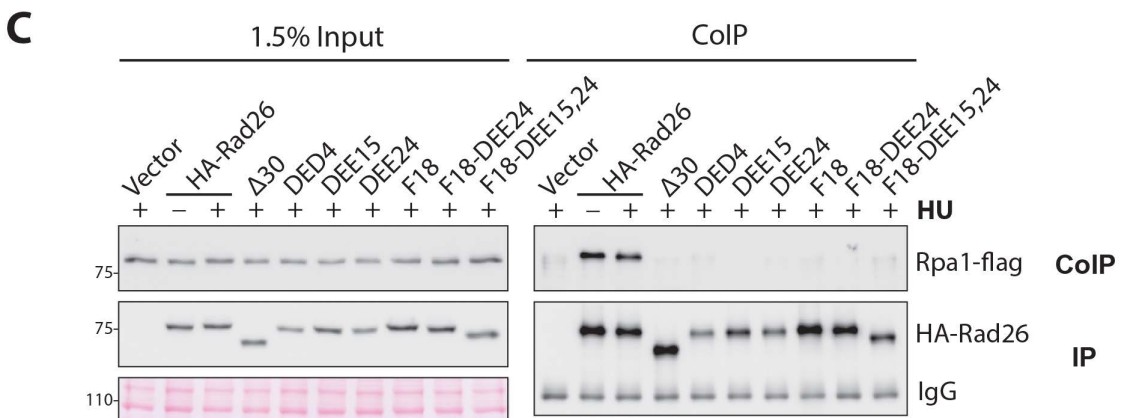

**Fig 2. The key residues in the RBD of Rad26 that bind to Ssb1. (A)** Wild-type and the Δ*rad26* cells carrying an empty vector or the vector expressing full-length Rad26 and the indicated mutants were examined by spot assay as in Fig 1C. Δ30 is the mutant lacking the first 30 residues. DED4, DEE15, and DEE24 are the simultaneous alanine substitutions of the triple acidic residues in each of the three motifs shown in Fig 1A. F18 is the alanine substitution of the residue in the DEE15 motif. F18-DEE24 and F18-DEE15,24 are the combined alanine substitutions. The dashed line indicates discontinuity. **(B)** Rad26 levels in the indicated cells used in A were examined by Western blotting as in Fig 1B (top panel). The asterisk indicates a cross-reacting material. A section of the stained membrane is shown at the bottom as the loading control. **(C)** The RPA-binding activity is lost in all mutants tested in A. The RPA binding activity in cells expressing full-length Rad26 or the indicated mutant was examined by co-IP as in Fig 1E. The left three panels represent 1.5% inputs.

budding yeast and other model systems [19–21,32–34]. This suggests that although the Rad26 binding to RPA is highly conserved in eukaryotes, its role in checkpoint initiation may not be as critical as previously thought. Alternatively, a redundant factor remains to be identified for promoting Rad3 signalling. In principle, checkpoint initiation should be a highly

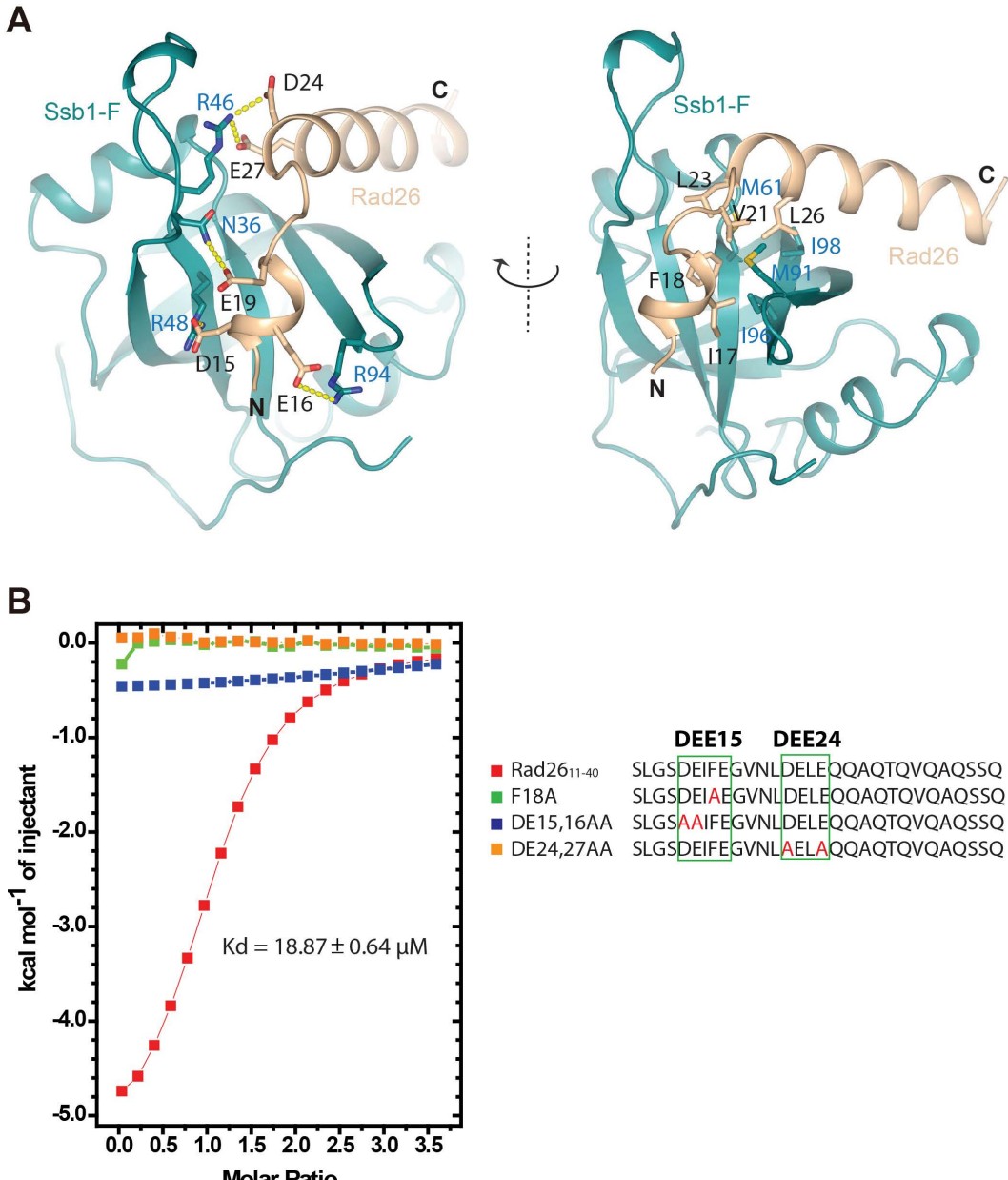

**Fig 3. Interdependent polar and hydrophobic interactions between Rad26-RBD and Ssb1-F domain. (A)** The protein sequences of Ssb1-F (1-122 aa) and Rad26 N-terminus (1-40 aa) were subjected to Alphafold3 modelling. The Ssb1-Rad26 complex structures were analysed in PyMOL, in which Ssb1-F is shown in teal, and the Rad26 N-terminus is in beige. The significant charge-charge interactions are shown on the left, whereas the hydrophobic interactions are shown on the right. **(B)** Four Rad26 (11-40 aa) peptides were chemically synthesized: WT (red squares), F18A (green squares), DE15,16AA (Blue squares), and DE24,27AA (orange squares) for measuring their binding to the purified Ssb1-F (1-122 aa) by calorimetry at 25 ˚C. Each titration was carried out with 19 injections, spaced in 150 seconds. The acquired calorimetric titration data were analyzed with Origin 7.0 software using the 'One Set of Binding Sites' fitting model for calculating the dissociation constant (Kd = 18.87 ± 0.64 µM for the WT peptide). The alanine substitutions in the highlighted DEE15 and DEE24 motifs of the Rad26 peptides are shown in red (right).

regulated process that relies on multiple, not a single cue, as such a mechanism would make noise control difficult. The constitutive binding of Rad26 with RPA, particularly at the fork, where RPA is enriched, may generate high background noise. In support of this notion, a threshold mechanism has been proposed in budding yeast, which may control such noise during the unperturbed S phase [35]. Alternatively, the constitutive binding of Rad26 with RPA only occurs in the cell lysate (Fig 1E), not in the chromatin fraction, where the chromatin-bound RPA may allow more dynamic interaction with Rad26 [36] and thus, the trenchant checkpoint initiation.

**A minor role of the KKRK motif in the checkpoint**. To identify the redundant factor that may contribute to Rad3$^{ATR}$ kinase signalling, we examined the KKRK motif in Rad26. This motif is conserved in budding yeast Ddc2, not in ATRIP, which has been shown to bind DNA [37,38] and is required for Mec1$^{ATR}$ kinase signalling in budding yeast [39,40]. Rad26 has five basic residues in this motif (Fig BA in S2 File), and we substituted each of them with alanine. The results showed that, unlike the budding yeast, none of the substitutions, whether in single or in combinations, significantly sensitized the cells to HU and MMS, except for the simultaneous mutation of $^{203}$KKRK$^{206}$, which moderately sensitized the cells to HU (Fig 4A). Western blotting showed that the levels of all mutant proteins were comparable to wild-type Rad26 (Fig 4B). Consistent with the drug resistance, the checkpoint signalling remained robust in MMS and moderately affected in HU (Fig 4C). These results showed that the KKRK motif plays a minor role or functions redundantly with other factors in the checkpoint. Furthermore, since the KKRK motif may serve as a potential nuclear localization signal (NLS) [41], this result shows that mutations of this motif do not affect the function of Rad26 in the nucleus (see below).

**Extrachromosomal expression is more reliable for the mutational analysis.** The results described above show that eliminating the RBD in Rad26 did not significantly reduce the Rad3 signalling. These experiments, however, relied on extrachromosomal expression, which increases the protein levels at least 2 times higher than the wild-type level (see below). This raises a concern of overexpression, which may enhance Rad3 kinase signalling in the Rad26 mutants. To address this issue, we integrated the Rad26 mutants at the genomic locus using the method illustrated in Fig IA in S2 File. As a control, wild-type *rad26* was integrated by the same method. All integrants were confirmed by colony PCR and Western blotting that detects the N-terminal HA epitope. Spot assay showed that the integrants of Δ10 and Δ20 mutations behaved like those expressed from the vector (compare Fig 1C with the top half of Fig IB in S2 File), whereas the Δ30, Δ50, and Δ70 integrants showed slightly increased drug sensitivities. When the substitution mutations were integrated, some mutants, such as DEE24, showed a dramatically increased drug sensitivity (compare Fig 2A with the middle part of Fig IB in S2 File). We then examined the protein levels. Since Rad26 levels in the integrants are low, Rad26 was IPed before Western blotting. We found that the protein levels of Δ30, Δ50, and Δ70 mutants were significantly lower than wild-type Rad26 (Fig IC in S2 File, left panels), and the integrated mutations of DEE24 and DEE15,24 almost eliminated the protein (Fig IC in S2 File, right panels). These results show that when integrated, some mutants showed heightened drug sensitivity due to lower protein levels. Consistent with this notion, integrants of the KKRK mutations behaved like those expressed from the vector, as the integration did not reduce the protein levels much (compare Fig 4 with the lower part of Fig IB in S2 File, and the right panels of Fig IC in S2 Files).

To provide further evidence, we expressed wild-type and mutant Rad26 from a vector in their respective integrants. As shown in Fig JA in S2 File, the wild-type integrant with extrachromosomal expression of wild-type Rad26 behaved almost the same as the empty vector control, although the protein level was 8-fold higher (Fig JB in S2 File). When phosphorylation of Mrc1 (Fig JC in S2 File) and Chk1 (Fig JD in S2 File) was examined, the overexpression did not affect the phosphorylation much in the wild-type Rad26 integrant, suggesting that the checkpoint is quite tolerant of the overexpression. We then expressed Rad26 deletion mutants from the vector in their respective integrants. Western blotting confirmed their overexpression of ~5–10-fold higher than in wild-type cells (Fig JB in S2 File). Spot assay showed that while the overexpression had no effect in Δ10 and Δ20, it increased the drug resistance in Δ30, Δ50, and Δ70 integrants (Fig JA in S2 File). As a result, phosphorylation of Mrc1 (Fig JC in S2 File) and Chk1 (Fig JD in S2 File) was also increased. We also expressed substitution mutants from a vector in their respective integrants (Fig K in S2 File), and similar rescuing

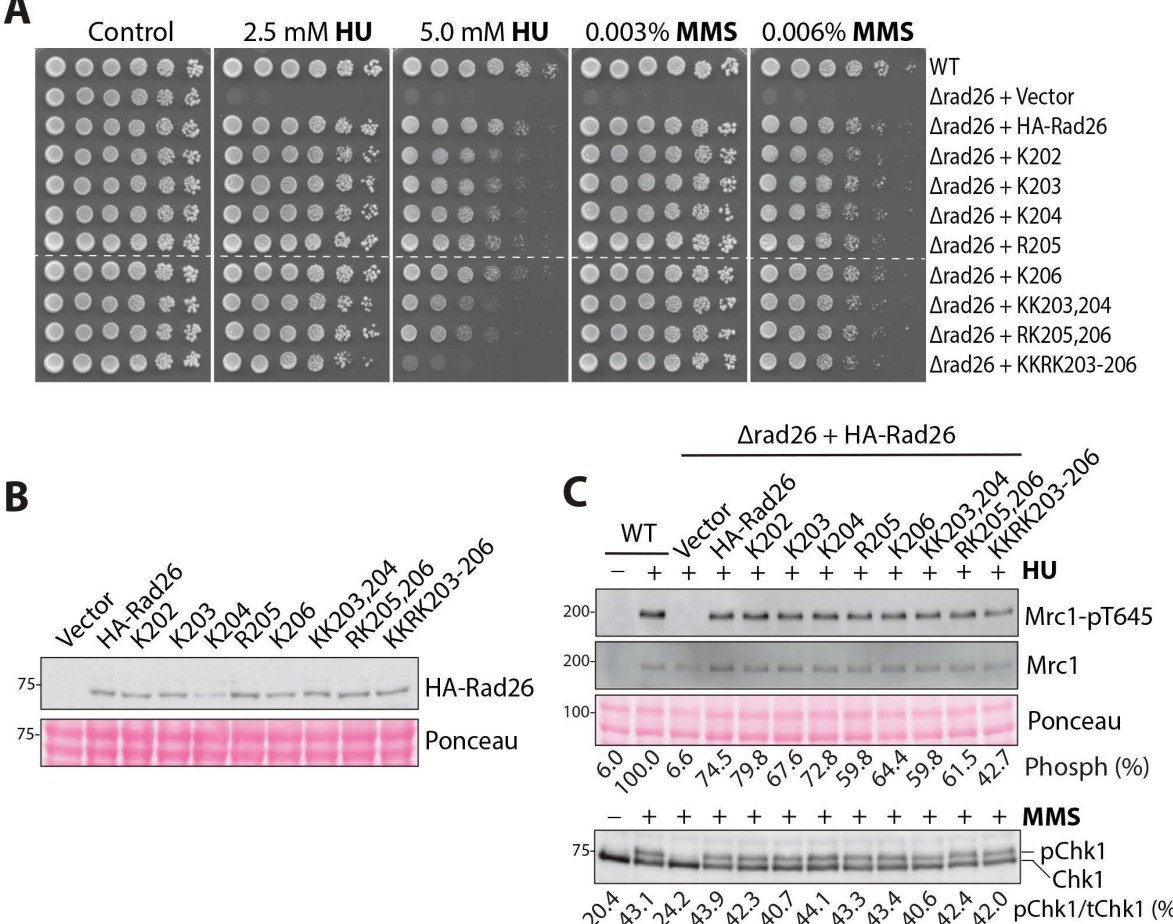

**Fig 4. Mutations of the KKRK motif in Rad26 cause minor checkpoint defects. (A)** Drug sensitivities of wild-type and the Δ*rad26* cells carrying an empty vector or the vector expressing full-length Rad26 or Rad26 with alanine substitutions of the indicated residues in the KKRK motif were examined by spot assay. Dashed line: discontinuity. **(B)** Rad26 levels were analyzed by Western blotting in the cells shown in A using α-HA antibody. **(C)** Rad3 phosphorylation of Mrc1 (upper two panels) and Chk1 (bottom panel) in the cells shown in A was examined as in Fig 1D. KK203,204 and KR205,206 are the double mutants, whereas the KKRK203-206 are a simultaneous mutation of all four basic residues.

effects were observed in the integrants of DEE24 and DEE15,24. The protein levels in the integrants of F18, F18-DEE24, and F18-DEE15,24 were less reduced or similar to wild-type cells and, therefore, their overexpression did not or mildly rescued the drug sensitivities and the phosphorylation defect of Mrc1 (Fig KC in S2 File) and Chk1 (Fig KD in S2 File) in the integrated mutants. As expected, extrachromosomal expression of the KKRK mutants had a minimal effect on the integrated mutations in drug sensitivity (Fig LA in S2 File) as well as the phosphorylation of Mrc1 (Fig LB in S2 File) and Chk1 (Fig LC in S2 File). Together, we conclude that integrating *rad26* mutations at the genomic locus can significantly reduce the protein levels in some, but not all, mutants, which causes exaggerated checkpoint defects and complicates the studies. Extrachromosomal expression of Rad26 is well tolerated by the checkpoint and thus serves as a reliable means for the mutational analysis.

**Combined mutations of the RBD and the KKRK motif almost eliminate Rad3 signalling at the fork.** Since the RBD deletion moderately affects Rad3 signalling, we reasoned that a functionally redundant factor may remain to be uncovered. To investigate this, we combined the mutations of KKRK with the Δ30 or F18-DEE15,24 mutations. To our

surprise, although the MMS sensitivity remained moderate, the combined mutations significantly sensitized the cells to HU more than the individual mutants (Fig 5A). The protein levels of the combined mutants were comparable to the single mutants and wild-type cells (Fig 5B). When Mrc1 phosphorylation was monitored, we found that the two combined mutants showed a significantly reduced phosphorylation (Fig 5C, upper panels). In the presence of MMS, although further reduced, a moderate level of Chk1 phosphorylation was observed in the combined mutants (Fig 5C, lower panel). We also made the combined mutations of Δ30 or F18-DEE15,24 with either KK or RK mutations, assessed their drug sensitivities (FigMA in S2 File), confirmed their expression levels (Fig MB in S2 File), and the checkpoint signalling defect in the DRC and DDC pathways (Fig MC in S2 File). Similar results were observed as in the combined mutant containing the KKRK mutation (compare Fig 5A–5C with Fig MA-MC in S2 File). To further investigate, we integrated the combined mutations containing KKRK, KK, or RK mutations at the genomic locus (Fig NA and NB in S2 File). Unfortunately, Rad26 levels were low in these integrants (Fig NC and ND in S2 File), which likely explains the further increased MMS sensitivity (compare Fig 5A with Fig NA in S2 File). However, the integrants of Δ30+KK and Δ30+RK showed nearly wild-type levels of Rad26 and behaved like the extrachromosomal-expressed mutants (Fig ND in S2 File, compare Fig NB in S2 File with Fig MA in S2 File) (see below)

**The combined mutations do not affect the nuclear localization of Rad26.** NLSs are commonly found in nuclear proteins that direct the cargo proteins to be imported into the nucleus. As mentioned above, the KKRK motif may serve as an NLS for Rad26. Although the mutations of this motif did not affect the nuclear function of Rad26 (Fig 4), the combined mutations of the KKRK and the RBD may cause mislocalization of Rad26, leading to the significant DRC defect. To address this issue, we separated the whole cell lysate into cytoplasmic and nuclear fractions [39,42], which were then analysed by Western blotting to detect the nuclear pore complex protein (NUP), the cytosolic enzyme glucose-6-phosphate dehydrogenase (G6PD), and Rad26, in comparison to the whole-cell lysate (Fig 5D). The results clearly showed that the two combined mutations did not affect the nuclear translocation of Rad26, suggesting the observed DRC defect is due to the mutations, not the mislocalization. Together, these results show that the KKRK motif cooperates with the RBD in initiating the Rad3 kinase signalling at the HU-treated fork. The subtle difference in MMS sensitivities between the single and combined mutants suggests that Rad3 signaling at the DNA damage site, which primarily occurs at G2 in MMS-treated cells, depends less on the cooperation (see Discussion).

**Integrant of the combined mutation of F18+KKRK showed significantly increased HU sensitivity.** Although integration of the majority of the RBD mutations decreased the protein level, we noticed that the Rad26 levels in the integrants of Δ30 and the combined mutations of Δ30 with KK or RK mutations were only slightly lower than those of the wild-type cells and the two integrated mutants behaved like the extrachromosomally expressed mutants with increased sensitivity to HU, not MMS. Since the mutations of F18 and KKRK did not significantly affect the protein levels (Figs 2B and 4B) and the F18 mutation eliminated the Rad26 binding to Ssb1 (Figs 2C and 3B), we then combined the F18 with the KKRK mutation and integrated the combined mutation at the genomic locus. The results showed that when integrated, the combined mutation of F18+KKRK significantly increased the HU sensitivity, whereas the MMS sensitivity was mildly increased (Fig 5G). Western blotting showed that Rad26 levels in the integrants of F18, KKRK, and F18+KKRK are almost the same as in wild-type cells (Fig 5F), showing that the increased HU sensitivity is not due to the reduced protein levels. Consistent with the increased HU sensitivity, Mrc1 phosphorylation in the combined mutant was significantly reduced in HU (Fig 5G, upper panels). In the presence of MMS, Chk1 phosphorylation was moderately reduced (Fig 5G, bottom panel). Together, these results eliminate the concerns associated with extrachromosomal expression and strongly support our conclusion that the RBD and the KKRK motif cooperate at the HU-treated fork to initiate Rad3 kinase signaling.

**The isolated KKRK motif binds weakly to DNA.** Previous studies have shown that the KKRK motif promotes DNA binding of budding yeast Ddc2 [37,38]. To determine whether this motif facilitates DNA binding in Rad26, we performed the DNA pull-down assay as previously described [37]. We found that wild-type Rad26 was robustly pulled down by dsDNA, confirming the previous result (Fig OA in S2 File). Western blotting showed that Ssb1 was also pulled down in the

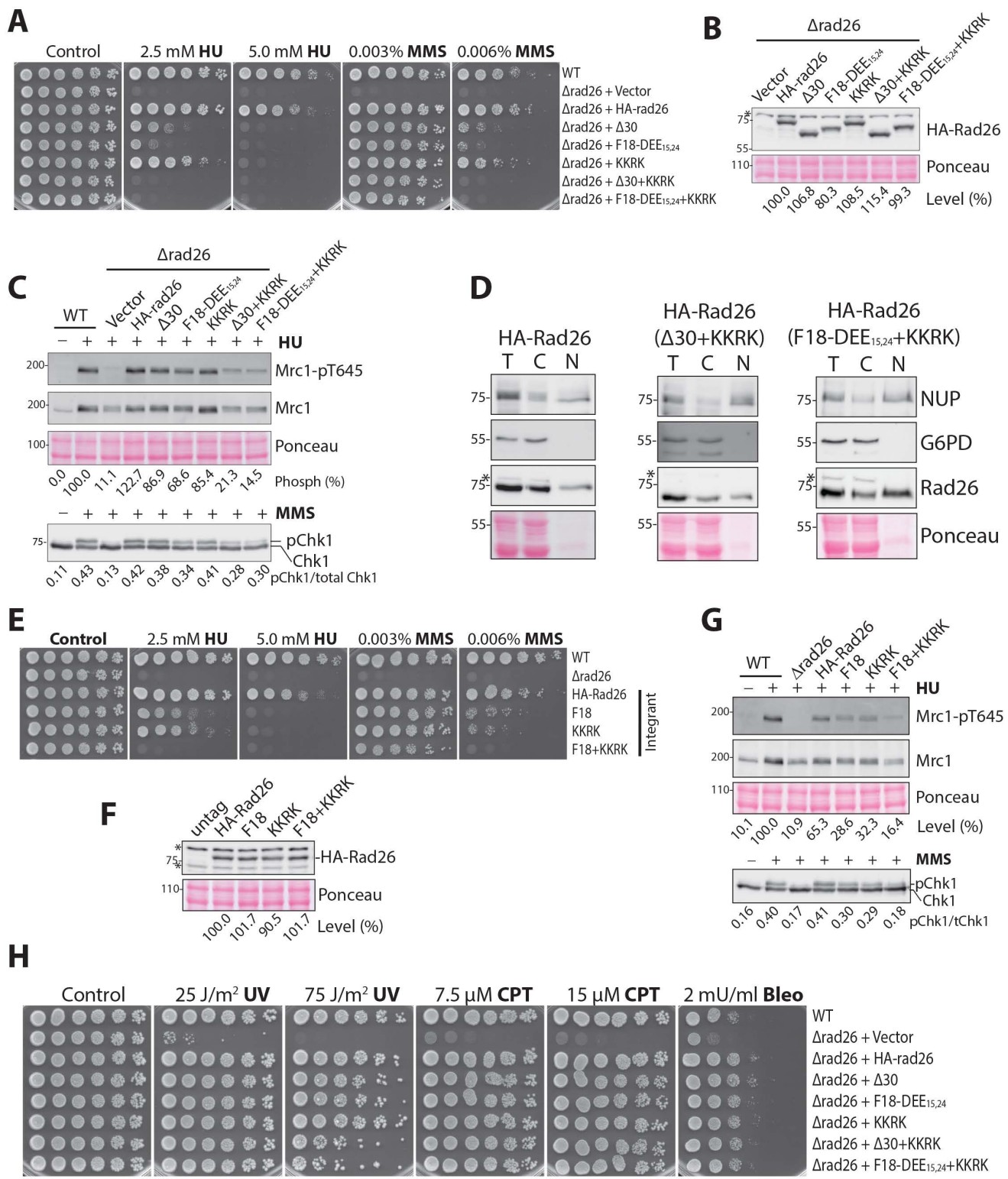

**Fig 5. Combined mutations of the RBD and the KKRK motif in Rad26 significantly affect the DRC, not the DDC pathway. (A)** Drug sensitivities of wild-type and the Δ*rad26* cells with the vector expressing full-length Rad26 or Rad26 with the indicated mutations were examined as in Fig 1C. Δ30+KKRK and F18-DEE15,24+KKRK are the combined mutants of the RBD and the KKRK motif. **(B)** Rad26 protein levels in the cells shown in A

PLOS Genetics

were examined as described in Fig 1B. Quantitation results are shown at the bottom. **(C)** Phosphorylation of Mrc1 (top three panels) and Chk1 (bottom panel) was examined as in Fig 1D. Quantitation results are shown underneath each panel. **(D)** The combined mutations do not affect the translocation of Rad26 to the nucleus. The Δ*rad26* cells expressing wild-type Rad26 (left four panels) and the Rad26 with the two combined mutations of the RBD and the KKRK motifs (middle and right four panels) were harvested to make the whole-cell lysates, which were subject to ultracentrifugation to separate the nuclei from the cytosolic fraction [39,42]. Total lysate **(T)**, cytosolic fraction **(C)**, and nuclear fraction **(N)** were analysed by Western blotting using antibodies against NUP, G6PD, and HA for Rad26. **(E)** Drug sensitivities of wild-type and the integrants of the indicated mutations were examined by spot assay. **(F)** Rad26 levels in the strain used in E were examined by Western blotting using α-HA antibody. **(G)** Phosphorylation of Mrc1 and Chk1 in the integrants used in E was examined and quantified as in C. **(H)** Drug sensitivities of wild-type and Δ*rad26* cells expressing Rad26 and Rad26 with the indicated mutations were examined by spot assay after treatment with UV radiation or in the presence of CPT or bleomycin.

same samples, which is not surprising and likely due to the end-resection of the dsDNA or a minimal amount of ssDNA bound to the magnetic beads. Since the Ssb1 pull-down was readily detected in every sample, we used it as the control (Fig OB in S2 File). Although the Rad26 levels in the cell extracts from Δ30 mutants were lower, elimination of the RBD slightly reduced the Rad26 pull-down as compared with Ssb1, suggesting that binding to RPA contributes to the Rad26 pull-down. To our surprise, mutation of the KKRK motif did not affect the Rad26 pull-down (Fig OA and OB in S2 File). AlphaFold modelling of Rad26 with either ssDNA or dsDNA shows that the KKRK motif does not make direct contact with DNA (Fig OC and OD in S2 File). We then purified GST-tagged Rad26(185–211aa) containing the KKRK motif and examined the DNA-binding activity by fluorescence anisotropy. In the presence of poly dT24, the recombinant protein showed the DNA-binding activity only when the protein concentration was increased to the 10–40 μM range (Fig OE in S2 File). In the presence of a mixed sequence of 25 bp dsDNA, an even weaker DNA binding activity was observed (Fig OF in S2 File). Under similar conditions, mutation of the KKRK motif eliminated the DNA-binding activity. This result shows that the KKRK motif, when isolated from Rad26 protein, has a weak affinity to DNA, particularly ssDNA. However, this activity likely contributes minimally to the DNA binding of Rad26.

**Resistance of the Rad26 mutants to DNA damage.** The Rad26 mutants with combined mutations of the RBD and the KKRK motif are less sensitive to MMS as compared to HU (Figs 1C, 2A, and 5A), suggesting a less effect on Rad3 signalling at the DNA damage sites (Fig 5C). To further investigate, we examined the sensitivity of the mutants to ultraviolet (UV) irradiation, camptothecin (CPT), and bleomycin (Fig 5H). While UV directly generates pyrimidine dimers in DNA [43], CPT stabilizes the covalent DNA-topoisomerase I complex, which, when encountered with the replication fork, can be converted into one-ended DSB [44]. Bleomycin cleaves DNA, generating SSB or DSB [45]. Consistent with the checkpoint function of Rad26 in the DDC, Δ*rad26* cells are sensitive to all the genotoxins tested (Fig 5H). Like the MMS treatment, most of the mutants are quite resistant to UV, except the combined mutants, which showed a moderately heightened sensitivity (compare Fig 5A with 5H). Remarkably, all mutants are quite resistant to CPT and bleomycin as compared with the cells expressing wild-type Rad26. This result shows that while simultaneous mutations of the RBD and the KKRK motif in Rad26 can dramatically attenuate Rad3 signalling at the fork, the signalling is less affected at the DNA damage site, particularly at the strand breaks.

## Discussion

To better understand the checkpoint initiation function of Rad26ATRIP, we used an integrated approach of genetic, biochemical, and structural modelling, focusing on the N-terminal region. Our *in vivo* and *in vitro* studies identified an RBD in the Rad26 N-terminus and the key residues in the RBD that bind to the F-domain of Ssb1, the large subunit of RPA in *S. pombe*. These residues interact extensively with the positively charged and hydrophobic residues in the Ssb1-F domain, and the polar and hydrophobic interactions are interdependent for Rad26 binding to Ssb1. However, deleting the RBD or mutating the key residues in the RBD abolishes the binding but results in only a moderate checkpoint defect (Figs 1 and 2). Although this outcome does not align with the current model, similar observations have been reported in other model organisms [20,21,32,34]. We also found that mutations of the KKRK motif in Rad26, a DNA-binding motif conserved in

budding yeast Ddc2, have a minor effect on the checkpoint. However, combining the mutations in the RBD and the KKRK motif nearly eliminated Rad3 signalling at the HU-treated fork and greatly increased cellular sensitivity to this agent. These findings indicate that the RBD and KKRK motif cooperate for Rad3 signalling at the fork. Although mutations of the KKRK motif appear to impact the checkpoint more than that of the RBD in Ddc2 [32,37,39], this cooperative mechanism is likely conserved in budding yeast. As mentioned above, the KKRK motif is not conserved in ATRIP, but human ATR-ATRIP can also bind DNA or damaged DNA independently of RPA [46,47], suggesting that the cooperative mechanism is also conserved in higher eukaryotes, which, if confirmed, may reconcile the previous controversial observations in budding yeast and other model systems [20,21,32,34]. Furthermore, although Rad3 kinase signaling is reduced in MMS-treated cells, cell survival was less impacted in MMS or UV as compared to HU (Figs 1C, 2A, and 5H). Importantly, the mutations minimally affected cell survival in CPT and bleomycin (Fig 5H), indicating that checkpoint initiation at DNA damage sites likely involves a different mechanism, especially for strand breaks.

**Extrachromosomal expression**. Rad26 is not required for cell growth, and its extrachromosomal expression can fully rescue the *rad26* null mutant. These technical benefits allow us to successfully identify the RBD in Rad26 and the key residues that bind to the Ssb1-F domain. However, we were initially concerned about overexpression and therefore integrated wild-type *rad26* and its mutants at the genomic locus. Unexpectedly, the integration reduced the protein levels of some but not all mutants, such as Δ30, Δ50, Δ70, DEE24, DEE15,24, and F18-DEE15,24 mutants (Fig IC in S2 File), which complicates the mutational analysis. The extrachromosomal expression generally increased Rad26 protein levels 2- to 12-fold of the wild-type level, depending on the mutations (Figs JB and KB in S2 File). However, the checkpoint is quite tolerant to Rad26 overexpression in *S. pombe* (Figs J and K in S2 File). One explanation is that the checkpoint function of Rad26 must be realized through Rad3, and the number of Rad3 molecules in a single cell is relatively small [48]. The reason why the protein levels are significantly lower in some mutants, but not others, remains unclear, although most of the mutations affect the binding to Ssb1. Nonetheless, extrachromosomal expression can overcome the complications caused by the low protein levels in some integrated Rad26 mutants, thereby offering a reliable means for the mutational analysis. Furthermore, the integrants of F18, KKRK, and the combined F18A+KKRK mutations show wild-type protein levels (Fig 5F). In the presence of HU, the F18 + KKRK integrant showed a significantly increased sensitivity and the defect in Mrc1 phosphorylation (Fig 5E and 5G). When treated with MMS, the F18 + KKRK integrant showed a mildly increased drug sensitivity and the Chk1 phosphorylation defect (Fig 5E and 5G). These results are exactly what is anticipated based on the extrachromosomal expression analysis.

**Interdependent interactions.** Several key residues in the RBD, including the 2nd and 3rd DEE and the F18 residue, are required for the binding to RPA. AlphaFold3 prediction and the biochemical studies confirmed their direct interactions with the Ssb1-F domain. These residues and those on the Ssb1-F form multiple polar and hydrophobic interactions. Unexpectedly, the *in vivo* and *in vitro* data strongly suggest that these interactions are interdependent for the binding. Although it is unclear why the contacts at the interfaces of Rad26-RBD and Ssb1-F have evolved to be interdependent, such an unusual binding mode may allow noise control by eliminating the weaker binding of fewer contacts. Since the binding is constitutive, it may generate spurious checkpoint signals, particularly at the replication forks where RPA is locally enriched. Further binding to ssDNA via the weak interaction with the KKRK motif or other parts of Rad26-Rad3 may therefore suppress the noise and allow trenchant checkpoint signalling to be initiated at the fork. This multi-factor cooperation may also explain why it is difficult to identify a checkpoint mutant in *ssb1* and *rfa1* [17,18]. In line with this notion, AlphaFold3 prediction showed that the mutated residues in *ssb1–1*(K33E) and *ssb1–7* (R11C-L69S) in the F domain (Fig A in S2 File) are not involved in the binding to Rad26-RBD (Fig 3A), similar to the mutated K45 residue in *rfa1-t11*, which explains why Ssb1 is co-IPed with Rad26 in the two mutants (Fig A in S2 File). In budding yeast, the binding of Ddc2-Mec1 to ssDNA may directly facilitate the activation of Mec1 without the need for the Mec1-activators Ddc1 and Dbp11 [38,49]. Whether this mechanism is conserved in fission yeast requires further studies. Also, the essential function of the 9-1-1 complex in the DRC remains incompletely understood, although the complex is not required for Rad3 phosphorylation of Mrc1 at the fork [27].

**The KKRK motif.** As mentioned above, the KKRK motif is conserved in Ddc2 and Rad26, not in ATRIP. The nature of the continuous stretch of basic residues makes this motif a perfect monopartite NLS of the nuclear proteins [41]. However, the results from this and a previous study in budding yeast Ddc2 [39] show that this motif is not required for the nuclear transport of Rad26. Rad26 is likely transported into the nucleus through another type of NLS or as a complex with Rad3. The constitutive binding with other nuclear proteins, such as RPA and Rad3, may retain Rad26 within the nucleus, preventing its mislocalization. Studies in budding yeast have shown that the KKRK motif binds DNA, which promotes its checkpoint activities of Ddc2 [37–40]. Our results, although preliminary, indicate that when isolated from the protein, the KKRK motif exhibits only a weak affinity for DNA, likely due to charge-charge interactions. However, unlike in budding yeast, this activity of the KKRK motif does not contribute significantly to the DNA-binding activity of Rad26 or the Rad26-Rad3 complex, which is consistent with its minimal checkpoint role observed in Fig 4. An earlier study showed that purified ATRIP binds ssDNA only in the presence of RPA [4]. This suggests that the DNA-binding activity of the KKRK motif observed in Ddc2 [37] is not highly conserved in eukaryotes, and the weak DNA-binding activity of the KKRK motif may play an important role in the DRC only when the RBD is eliminated from Rad26 in fission yeast. Alternatively, this motif has another yet unknown function that cooperates with the RBD for checkpoint initiation at the fork. Nonetheless, the cooperation of the KKRK motif and the RBD uncovered here may facilitate future biochemical studies on the recruitment and subsequent activation of Rad26-Rad3 in the DRC pathway.

**Checkpoint initiation at DNA damage sites.** Although the simultaneous mutations of the RBD and the KKRK motif significantly affect the DRC (Fig 5A and 5C), the mutants show only a moderate sensitivity to MMS and UV and a minimal sensitivity to CPT and bleomycin (Fig 5A and 5H). The Chk1 phosphorylation is reduced to ~50–60% of the wild-type level in the mutants treated with MMS. Since MMS and UV also generate replication stress and their cellular resistance depends in part on the DRC, this, together with the moderately reduced Chk1 phosphorylation, likely explains the moderate sensitivity to the two agents. RPA and ssDNA may be more abundant at the fork than at the DNA damage site, which makes the DRC more dependent on the binding to RPA and ssDNA. However, the moderate and the minimal sensitivity to various DNA-damaging agents strongly suggest that initiation of the DDC likely involves a different mechanism that depends less on the cooperation of the two functional units of Rad26. Consistent with this notion, several DNA repair proteins have been reported in other model systems to facilitate the recruitment of the checkpoint sensor proteins to the damage sites for checkpoint initiation [50–53]. Identification of those factors in a genetically tractable model system, such as *S. pombe,* would be helpful to fully understand the checkpoint initiation mechanisms.

## Materials and methods

**Yeast strains and plasmids**. The *S. pombe* strains were cultured at 30°C in YE6S (0.5% yeast extract, 3% dextrose, and 6 supplements) or synthetic EMM6S medium lacking the appropriate supplement [54]. Yeast strains, plasmids, and PCR primers used in this study are listed in Tables A, B, and C in S1 File, respectively. Mutations were generated by PCR using Phusion DNA polymerase (New England Biolabs) and confirmed by DNA sequencing (Retrogen, San Diego, CA). The cellular effects of Rad26 mutants expressed on a vector in Δ*rad26* cells are summarized in Table D in S1 File.

**Spot assay and integration of *rad26* mutations**. Sensitivities to HU and the DNA-damaging agents MMS, UV, CPT, and bleomycin were determined by standard spot assay as described previously [55]. For integration, the *rad26* expression cassette with the *rad26* promoter and *nmt1* terminator linked with a kanR gene and the *rad26* 3' UTR was digested with PshAI and PvuII for purification of a 5.7 kb fragment and subsequent transformation into a Δ*rad26::ura4* strain (Fig I in S2 File). Yeast colonies that lost the *ura4* marker and gained resistance to G418 were selected for colony PCR and Western analysis using α-HA antibody to confirm the integration.

**IP and co-IP.** $1 \times 10^8$ logarithmically growing cells were harvested and saved at -20°C. The frozen cell pellets were lysed by mini-bead beater in the buffer containing 25 mM HEPES/NaOH, pH 7.5, 50 mM NaF, 1 mM NaVO$_4$, 10 mM NaP$_2$O$_7$, 40 mM ß-glycerophophate, 0.1% Tween 20, 0.5% NP-40, and protease inhibitors. The lysates were

centrifuged at 16,000 g, 4˚C, for 5 min to make the cell extract. Agarose resins linked with α-HA or α-Myc antibodies (Santa Cruz) were washed three times with Tris-buffered saline containing 0.05% Tween 20 (TBS-T), and then incubated with the cell extract by rotating for 2 h at 4˚C. The resins were washed with TBS-T at 4˚C for 20 min, repeated three times. The IPed samples were separated by 8% SDS-PAGE followed by Western blotting. For co-IP of HA-Rad26 with Ssb1, 3 μl Protein G Dynabeads (Invitrogen) prewashed three times in PBS-T was mixed with 1 μg α-HA or α-Flag antibody (Stratagene) for 60 min and washed three times. The antibody-loaded Dynabeads were incubated with the cell lysate as described above for IP, except the beads were collected using a magnetic rack during the washes. The samples were separated by 8% SDS-PAGE followed by Western analysis. The blotting signal was detected by electrochemiluminescence using ChemiDoc XRS Imaging system (BioRad). Intensities of the bands were quantified and analysed by ImageLab (BioRad).

**Western analysis.** Rad3-dependent phosphorylation of Mrc1-T645 was analysed by Western blotting using the phospho-specific antibody described in previous studies [23,27,28]. Phosphorylation of Chk1 was examined by mobility shift assay [24]. Ssb1 was detected by either α-Flag antibody or the custom antibody against Ssb1 made by Cocalico Biologicals, Inc. [17]. Rad3 was tagged with 10xMyc epitope at the N-terminus and detected by Western blotting using α-Myc antibody.

**Purification of recombinant Ssb1-F domain and GST-Rad26 (185–211aa)** The DNA sequence of Ssb1-F(1–122 aa) domain was cloned into a modified pRSFDuet-1 vector (Novagen) with an N-terminal 6-His-sumo tag, using Hieff Clone Plus One Step Cloning Kit (Yeasen). The Rad26(185–211) DNA fragment was cloned into a modified pGEX-6P-1 vector (Novagen) with a GST tag. The plasmids were transformed into *E. coli* Rosetta 2 (DE3) cells (Novagen), which were grown in LB medium at 37 °C until the OD 600 reached 0.6–0.8. Protein overexpression was induced by the addition of 0.5 mM isopropyl β-D-thiogalactoside (IPTG), followed by incubation at 18 °C for 16 h. Cells were harvested by centrifugation, resuspended in lysis buffer (20 mM HEPES, 300 mM NaCl, 20 mM imidazole, 10% glycerol, 0.3 mM TCEP, pH 8.0), and lysed by a high-pressure homogenizer at 4 °C. The cell lysate was centrifuged at 12,000 rpm for 60 min to obtain a soluble extract. After nickel affinity pull-down, the 6-His-sumo tag of Ssb1-F was cleaved off by Ulp1 protease and removed by a second nickel column. The flow-through was then passed through a heparin column (Cytiva) and eluted with a gradient of 0-1.0 M NaCl in a buffer of 20 mM HEPES, pH 7.5, 10% glycerol, and 0.3 mM TCEP. Fractions containing Ssb1-N were pooled and concentrated, then further purified on a Superdex 200 increase gel filtration column (Cytiva) in a buffer containing 20 mM HEPES, pH 7.5, 200 mM NaCl, 0.3 mM TCEP. For GST-Rad26(185–211) purification, the clarified supernatant was loaded onto a GST gravity column pre-equilibrated with lysis buffer. The protein was eluted with a buffer containing 20 mM HEPES, pH 7.5, 500 mM NaCl, 10% glycerol, 0.3 mM TCEP, 0.1 mM PMSF, and 25 mM Reduced Glutathione (GSH). The eluted protein was concentrated and loaded onto a Superdex 200 Increase 10/300 column (Cytiva) with gel filtration buffer containing 20 mM HEPES, pH 7.5, 200 mM NaCl, 0.3 mM TCEP. The peak fractions of GST-Rad26 (185–211) were collected and concentrated, aliquoted, snap frozen in liquid nitrogen, and stored at -80˚C. The mutant protein of GST-Rad26 (185–211 K203A-K204A-R205A-K206A) was purified by following the same protocol.

**Isothermal titration calorimetry (ITC).** Four Rad26 (11–40 aa) peptides (WT, F18A, DE15,16AA, and DE24,27AA) were synthesized Ji'er Biochemical. All ITC titrations were carried out using a MicroCal PEAQ-ITC instrument (Malvern) at 25 °C with different peptides in the syringe and the purified Ssb1-F (1–122 aa) in the cell. Ssb1-F and peptide samples were dialyzed against a working buffer consisting of 20 mM HEPES, 100 mM NaCl, 0.3 mM TCEP, pH 7.5. Each titration was carried out with 19 injections, spaced 150 seconds apart, at a stir speed of 500 rpm. The acquired calorimetric titration data were analyzed with Origin 7.0 software using the 'One Set of Binding Sites' fitting model.

**Fluorescence-based DNA-binding assay.** All fluorescence polarization DNA-binding assays were performed on 96-well black polypropylene plates. 5 nM 5' FAM-labeled ssDNA (dT)24 or 25 bp dsDNA (GenScript) were incubated with increasing concentrations of WT or KKRK mutant GST-Rad26 (185–211) at room temperature for 10 min in a buffer containing 40 mM Tris-HCl, pH 8.0, 100 mM NaCl, 2 mM $MgCl_2$, and 0.3 mM TCEP. Fluorescence polarization (mP) was

measured at room temperature using a BioTek Synergy H1 reader (Agilent). The curves were fitted in GraphPad Prism v.10.0 using non-linear regression, one site-specific binding.

**DNA-pull-down assay.** The assay was conducted following the previously described method [37]. Briefly, a 72 bp dsDNA (0.25 µg) created by annealing biotin-P1 with P2 oligonucleotides was incubated with pre-equilibrated 10 µl streptavidin-coated magnetic Dynabeads at 25°C for 30 min. Whole cell extracts were prepared from 15.0 OD frozen cell pellets, clarified by centrifugation at 16,000 g, 4°C, for 5 min, and then incubated with the Dynabeads for 60 min at 4°C on a shaking platform. The magnetic beads were collected using a magnetic rack and washed three times in the buffer of 50 mM HEPES (pH 7.4), 0.1 M NaCl, 1% Triton X-100, 0.1% 2-mercaptothanol, and protease inhibitors. The samples were separated in an 8% SDS-PAGE followed by Western blotting using the α-HA antibody to reveal HA-Rad26. The blot was stripped and re-probed using α-Ssb1 antibody.

**Structure prediction by AlphaFold3**. The protein sequences of S. pombe Ssb1-F(1–122 aa) (Q92372) and Rad26 (1–40 aa) (P36632) or Rad26 dimer with ssDNA or dsDNA were submitted to the Alphafold3 server (https://alphafold-server.com/). The resulting models of the Ssb1-Rad26 complex structures were analyzed in PyMOL (The PyMOL Molecular Graphics System, Version 3.0, Schrödinger, LLC).

**Cytoplasmic and Nuclear fractionation**. *S. pombe* nucleus and cytoplasm were fractionated using the method described for *S. cerevisiae* with modifications [39,42]. The cells expressing HA-Rad26 from a vector were cultured overnight in EMM6S[leu-] medium at 30°C. When the OD600 reached 0.6 to 0.7, 30.0 OD cells were harvested by centrifugation. The cell pellet was resuspended in 1.5 ml CSE buffer (1.2 M D-sorbitol, 50 mM sodium citrate/phosphate, and 40 mM EDTA, pH 5.6) containing 37.5 mg Lysing enzyme (Sigma) and 19.0 mg Zymolase 20T (Seikagaku) in a 2 ml screw-cap tube. After incubation at 30°C for 40 min, the cell suspension was split into two parts and layered onto 0.75 ml 7.5% Ficoll (Fisher Scientific) in two tubes made in CSE buffer containing 1.0 mM PMSF. After centrifuging in a benchtop centrifuge at 6000 rpm, 4°C, for 5 min, the supernatant was removed. The pellet was resuspended in 2.0 ml Lysis buffer (20 mM $KH_2PO_4$, 1 mM $MgCl_2$, pH 6.5) containing 18% Ficoll and protease inhibitors (20 µg/ml leupeptin, 1.0 mM PMSF, 10 µg/ml pepstatin, 10 mM benzamidine, and 0.4% aprotinin). After being transferred to a 3 ml Dounce homogenizer and gently homogenized with 2 x 10 strokes in about 5–6 min on ice, the lysate was separated into two 1.5 ml tubes, centrifuged at 8000 g, 4°C, for 5 min. The supernatants were transferred to two new 1.5 ml tubes and centrifuged again at 8000 g, 4°C, for 10 min. The whole cell lysates were combined and layered onto 1.5 ml lysing buffer containing 30% Ficoll and 1 mM PMSF in a 5 ml thin-wall centrifuge tube (Fisher Scientific), and centrifuged in a MX120$^+$ micro-ultracentrifuge (Sorvall) with the S52-ST swing bucket rotor at 35,000 rpm, 4°C, for 60 min. A portion of the 18% Ficoll section was recovered as the cytoplasmic fraction, and the pellet was dissolved in 2.5x SDS loading buffer as the nuclear fraction. Samples of the two fractions and total lysate were analysed by SDS-PAGE followed by Western blotting using antibodies α-NUP (EMD Millipore), G6PD (Invitrogen), respectively. Rad26 was detected using α-HA antibody.

## Supporting information

**S1 File. Supplementary Tables.**
(PDF)

**S2 File. Supplementary Figures.**
(PDF)

## Acknowledgments

We thank NBRP/YGRC in Japan for the yeast strains. We also thank other members of the Xu lab and the Zhou lab for their help and critical reading of the manuscript.

## Author contributions

**Conceptualization:** Yong-jie Xu.

**Data curation:** Yong-jie Xu, Anmin Gao, Kamal Dev, Yuyuan Zheng, Mashael Y. Alyahya, Chun Zhou.

**Formal analysis:** Yong-jie Xu, Kamal Dev, Yuyuan Zheng.

**Funding acquisition:** Yong-jie Xu, Chun Zhou.

**Investigation:** Yong-jie Xu, Anmin Gao, Kamal Dev, Yuyuan Zheng, Mashael Y. Alyahya, Sairam Pasam, Guramrit Kaur, Chun Zhou.

**Methodology:** Yong-jie Xu.

**Project administration:** Yong-jie Xu, Chun Zhou.

**Resources:** Yong-jie Xu.

**Supervision:** Yong-jie Xu, Chun Zhou.

**Validation:** Yong-jie Xu.

**Writing – original draft:** Yong-jie Xu.

**Writing – review & editing:** Yong-jie Xu, Chun Zhou.

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
