## [Decision Letter · Decision Letter 0]

16 Nov 2025

PGENETICS-D-25-01080

The RPA-binding domain and the KKRK motif in Rad26ATRIP cooperate at the perturbed DNA replication fork for initiating checkpoint signalling

PLOS Genetics

Dear Dr. Yong-Jie Xu,

Thank you for submitting your manuscript to PLOS Genetics. Your manuscript has been reviewed by three referees who are positive about your study but have a number of specific comments that should be addressed.. Therefore, we invite you to submit a revised version of the manuscript that addresses the points raised by the reviewers.

Please submit your revised manuscript within by Dec 16 2025 11:59PM. If you will need more time than this to complete your revisions, please reply to this message or contact the journal office at plosgenetics@plos.org. Please include the following items when submitting your revised manuscript:

We look forward to receiving your revised manuscript.

Kind regards,

Michael Snyder

Academic Editor

PLOS Genetics

Pablo Wappner

Section Editor

PLOS Genetics

Aimée Dudley

Editor-in-Chief

PLOS Genetics

Anne Goriely

Editor-in-Chief

PLOS Genetics

**Journal Requirements:**

At this stage, the following Authors/Authors require contributions: Marshael Y. Alyahya. Please ensure that the full contributions of each author are acknowledged in the "Add/Edit/Remove Authors" section of our submission form.

The list of CRediT author contributions may be found here: https://journals.plos.org/plosgenetics/s/authorship#loc-author-contributions

- ® on page: 26.

2) If any authors received a salary from any of your funders, please state which authors and which funders..

6) Kindly revise your competing statement in the online submission form to align with the journal's style guidelines: 'The authors declare that there are no competing interests.'

**Reviewers' comments:**

Reviewer's Responses to Questions

**Comments to the Authors:**

Reviewer #1: This manuscript describes the examination of the fission yeast checkpoint protein Rad26 (Ddc2/ATRIP). The authors present data to suggest that Rad26 uses its RBD (RPA binding domain) to associate with Ssb1 (RPA), and its KKRK motif (the corresponding Ddc2 motif binds to DNA and is critical for checkpoint) to bind DNA. Extensive mutagenesis data from the authors suggest that these two interactions collaborate to enable the activation of the Rad3 (Mec1/ATR) kinase during DDR mostly when cells are treated with HU and MMS but not other types of genotoxins. Many of the examined mutants unfortunately reduced protein levels, making it difficult to reach solid conclusions. This issue was somewhat mitigated by using plasmid carrying Rad26 and its various mutants. The most effective mutations that largely maintained Rad26 protein level is the combination mutations of the two motifs, such as F18, DEE24, KKRR mutations, which increased the sensitivity to MMS and HU, and reduced DDR and DDC activation. Interestingly, the mutant is not sensitivity to other genotoxin including CPT, UV, and Bleomycin.

The authors concluded that the Rad26’s RPA binding domain and its putative DNA binding sites can promote checkpoint initiation under HU and MMS conditions. The experiments were well done, and the writing is mostly clear. I have a few suggestions to improve the study and the readability of the manuscript.

First , the large number of mutations examined can be difficult to track by readers. Perhaps the authors could provide a schematic to summarize the key effects for each category of mutants by listing one representative mutant per category.

Second, the authors provided a nice biochemical result to confirm the predicted RPA and Rad26 interaction. Can a similar test be done to confirm that the KKRK site in Rad26 is indeed involved in DNA binding?

Third, it is surprising that RBD and KKRK mutations when combined only resulted in sensitivity toward MMS and HU, but not CPT and Bleomycin. Can the authors determine whether one of the combined mutants affects the checkpoint activation in CPT or Bleomycin conditions?

Reviewer #2: In this study, Xu et al. present an insightful analysis of Rad26 function in replication checkpoint activation in Schizosaccharomyces pombe. Using a series of well-designed mutants, the authors uncover a cooperative mechanism between the RPA-binding domain (RBD) and a conserved KKRK motif of Rad26 in activating the replication checkpoint at perturbed DNA replication forks. They further demonstrate that this function is distinct from the DNA damage response signaling pathway. The experimental design is logical, the data are solid, and the conclusions are generally convincing. These findings provide new insights into the specific role of the N-terminus of Rad26 in initiating the Rad3/Rad26 checkpoint axis and suggest that replication checkpoint and DNA damage response signaling are distinctly regulated, depending on the underlying DNA structures and the factors that recognize them.

While the manuscript is strong overall, several points should be addressed prior to publication:

1. It is uncommon to abbreviate “DNA replication checkpoint (DRC)” or “DNA damage checkpoint (DDC).” Please use the full terms throughout the manuscript.

2. The last paragraph of the Introduction section (lines 113–128) largely reiterates the results. This redundancy is unnecessary for an Introduction-type paragraph. It can be substantially shortened and merged with the preceding paragraph ending at line 111.

3. Lines 275–277: Mutation of DED4 also abolished RPA binding of Rad26 as did DEE15 and DEE24; yet, the conclusion omitted DED4, which should be included here for completeness.

4. Lines 319–321: The authors propose an interesting alternative explanation for the observed differences in Rad26 function regarding RPA binding and checkpoint signaling through its N-terminus. Can this hypothesis be tested experimentally—for example, by assessing Rad26–RPA binding on chromatin in cells treated or untreated with HU or MMS?

5. In Figure 5A, the difference in cell viability between HU and MMS treatment for the combined mutation appears subtle. This variation might simply reflect the inherent differences between these two agents, rather than a specific effect on checkpoint or DNA damage response signaling. Please discuss on this point.

6. Regarding Figure 5E, although CPT can induce double-strand breaks (DSBs), it is not primarily a DSB-inducing agent. Moreover, the panel for bleomycin in Figure 5E appears blacked out on the right side, making it difficult to evaluate strain sensitivity. A clearer image should be provided.

7. Parts of the Discussion—particularly the section on the NLS—could be condensed. Since this topic was introduced in Introduction and Results sections and does not appear to be a major determinant, a concise discussion would be sufficient.

Reviewer #3: The works by Chun Zhou and colleagues investigated how the RPA binding domain and the KKRK motif, a putative DNA binding domain, of the checkpoint sensor Rad26, cooperate to initiate checkpoint signaling and resistance to genotoxic drugs. The interaction between the single stranded protein RPA and Rad26, the co-activator of the checkpoint senor kinase Rad3 (Human ATR) is thought to be instrumental in initiating checkpoint signaling. Previous reports from this group have screened for the identification of mutant of the large subunit of RPA defective for checkpoint activation. However, the mutants identified remain able to interact with Rad26. Therefore, the authors turned their attention to Rad26. The authors have conducted a careful and extensive structure-function analysis of the N-terminal part of Rad26, harboring a RPA Binding Domain (RBD), by combining biochemical and genetic analysis. First, the authors identified that the first 50 amino acids of Rad26 are required to promote Rad26 interaction with the large subunit of RPAs and checkpoint signaling. Then, several key residues (F18, DEE15 and DEE24) within the first 30 amino acids of Rad26, were shown to be critical for direct interaction with the large subunit of RPA in vitro, using ITC, revealing that Rad26-RPA binding involves a series of interdependent polar and hydrophobic interactions. When overexpressed, the corresponding Rad26 mutated forms resulted in sensitivity to Hydroxy urea (HU), reflecting a defect in the DNA replication checkpoint, and a very moderate sensitivity to methyl methane sulfonate (MMS), indicating a robust DNA Damage checkpoint (DDC). The analysis of Chk1 phosphorylation (DDC) and Mrc1 (DRC) further confirmed that the interaction between RPA and Rad26 can be compromised without affecting checkpoint signaling. Therefore, the authors looked for additional regulation layers and turned their attention to the KKRK motif of Rad26, a potential NLS and DNA binding domain. The substitution of all residues in alanine rendered cells sensitive to HU but not to MMS without affecting checkpoint signaling. Finally, the authors have combined mutations affecting both interaction with RPA and in the KKRK motifs to establish that these two domains cooperate to ensure checkpoint signaling. Moreover, the authors established that the KKRK motif is not required to promote the nuclear localization of Rad16. An interesting message of this work is that the interaction between Rad26 and RPA is constitutive (occurring in the absence of DNA damage) and that this interaction can be genetically uncoupled from checkpoint signaling. A second important message is the fact that checkpoint signaling requires distinct mechanisms in response to different genotoxic agents. However, there are two main limitations in this study (see major points). First, it would strengthen the message if the authors could provide evidence that the KKRK promote DNA binding. Secondly, since most of the work has been conducted using overexpression of Rad26 mutated forms, it limits the interpretation of the data. Nonetheless, the authors have done their best to conduct experiments using chromosomal integrated mutants, resulting in the alteration of the expression of Rad26 mutated forms, further complicating the interpretation of the data. Overall, the authors have conducted a robust and careful analysis of the role of the N-terminal part of Rad26 in promoting checkpoint signaling and resistance to genotoxic agents that deserves to be shared with a broad community working on genome stability.

Major points

1. The integrated mutated form Rad26-F18A rendered cells sensitive to HU and not to MMS without affecting the expression level of Rad26 (Figure S9) and checkpoint signaling (Figure S11). This single mutation impairs the interaction with RPA (Figure 3) in vitro. Does the integrated Rad26-F18A mutant affect interaction with RPA? This would strengthen the conclusion that RPA-Rad26 interaction is dispensable to initiate checkpoint signaling in more physiological conditions.

2. The integrated mutations in the KKRK motif (203-206) rendered cells sensitive to HU but not to MMS (Figure S12), without affecting the expression level of Rad26 (Figure S14) and checkpoint signaling (Figure S12). It would strengthen the conclusion if the authors could combine the F18A and mutations in the KKRK motif (203-206) and show that the integrated mutated form is indeed defective for checkpoint signaling, especially for the DRC and not the DDC.

3. Does the KKRK motif of Rad26 promote DNA binding? If so, is the DNA binding activity of Rad26 altered by mutation of the motif?

Minor points

3. Why the large subunit of RPA is sometimes named SSb1 and sometimes RPA1 ? I would suggest to keep the nomenclature consistent.

4. Figure S3 : Why Mrc1 is hardly visible in the western-blot. How its phosphorylation can be detected if the expression if down regulated ?

5. Line 196: “phosphor-specific antibody” Phospho-specific ?

6. Figure S9 is called after S10 and S11.

**Have all data underlying the figures and results presented in the manuscript been provided?**

Reviewer #1: Yes

Reviewer #2: Yes

Reviewer #3: Yes

PLOS authors have the option to publish the peer review history of their article (what does this mean? ). If published, this will include your full peer review and any attached files.

**Do you want your identity to be public for this peer review?** For information about this choice, including consent withdrawal, please see our Privacy Policy .

Reviewer #1: No

Reviewer #2: No

Reviewer #3: No

**Figure resubmission:**
---

## [Editor Report · Decision Letter 1]

9 Feb 2026

Dear Dr Xu,

We are pleased to inform you that your manuscript entitled "The RPA-binding domain and the KKRK motif in Rad26ATRIP cooperate at the perturbed DNA replication fork for initiating checkpoint signalling" has been editorially accepted for publication in PLOS Genetics. Congratulations!

Yours sincerely,

Michael Snyder

Academic Editor

PLOS Genetics

Pablo Wappner

Section Editor

PLOS Genetics

Aimée Dudley

Editor-in-Chief

PLOS Genetics

Anne Goriely

Editor-in-Chief

PLOS Genetics

BlueSky: @plos.bsky.social

Comments from the reviewers (if applicable):

**Data Deposition**

http://datadryad.org/submit?journalID=pgenetics&manu=PGENETICS-D-25-01080R1

**Press Queries**

---

## [Editor Report · Acceptance letter]

PGENETICS-D-25-01080R1

The RPA-binding domain and the KKRK motif in Rad26ATRIP cooperate at the perturbed DNA replication fork for initiating checkpoint signalling

Dear Dr Xu,

We are pleased to inform you that your manuscript entitled "The RPA-binding domain and the KKRK motif in Rad26ATRIP cooperate at the perturbed DNA replication fork for initiating checkpoint signalling" has been formally accepted for publication in PLOS Genetics! Your manuscript is now with our production department and you will be notified of the publication date in due course.

With kind regards,

Lilla Horvath

PLOS Genetics

On behalf of:
